# Study of HgCdTe (100) and HgCdTe (111)B Heterostructures Grown by MOCVD and Their Potential Application to APDs Operating in the IR Range up to 8 µm

**DOI:** 10.3390/s22030924

**Published:** 2022-01-25

**Authors:** Małgorzata Kopytko, Jan Sobieski, Waldemar Gawron, Piotr Martyniuk

**Affiliations:** 1Institute of Applied Physics, Military University of Technology, 2 Kaliskiego St., 00-908 Warsaw, Poland; jsobieski@vigo.com.pl (J.S.); wgawron@vigo.com.pl (W.G.); piotr.martyniuk@wat.edu.pl (P.M.); 2Vigo System S.A., 129/133 Poznańska St., 05-850 Ożarów Mazowiecki, Poland; 3Department of Electrical Engineering, The Ohio State University, 2024 Neil Avenue, Columbus, OH 43210, USA

**Keywords:** infrared detectors, avalanche photodiodes, avalanche multiplication, impact ionization, HgCdTe, avalanche gain, excess noise factor

## Abstract

The trend related to reach the high operating temperature condition (HOT, temperature, *T* > 190 K) achieved by thermoelectric (TE) coolers has been observed in infrared (IR) technology recently. That is directly related to the attempts to reduce the IR detector size, weight, and power dissipation (SWaP) conditions. The room temperature avalanche photodiodes technology is well developed in short IR range (SWIR) while devices operating in mid-wavelength (MWIR) and long-wavelength (LWIR) require cooling to suppress dark current due to the low energy bandgap. The paper presents research on the potential application of the HgCdTe (100) oriented and HgCdTe (111)B heterostructures grown by metal-organic chemical vapor deposition (MOCVD) on GaAs substrates for the design of avalanche photodiodes (APDs) operating in the IR range up to 8 µm and under 2-stage TE cooling (*T* = 230 K). While HgCdTe band structure with molar composition *x_Cd_* < 0.5 provides a very favorable hole-to-electron ionization coefficient ratio under avalanche conditions, resulting in increased gain without generating excess noise, the low level of background doping concentration and a low number of defects in the active layer is also required. HgCdTe (100) oriented layers exhibit better crystalline quality than HgCdTe (111)B grown on GaAs substrates, low dislocation density, and reduction of residual defects which contribute to a background doping within the range ~10^14^ cm^–3^. The fitting to the experimentally measured dark currents (at *T* = 230 K) of the N^+^-ν-p-P^+^ photodiodes commonly used as an APDs structure allowed to determine the material parameters. Experimentally extracted the mid-bandgap trap concentrations at the level of 2.5 × 10^14^ cm^−3^ and 1 × 10^15^ cm^−3^ for HgCdTe (100) and HgCdTe (111)B photodiode are reported respectively. HgCdTe (100) is better to provide high resistance, and consequently sufficient strength and uniform electric field distribution, as well as to avoid the tunneling current contribution at higher bias, which is a key issue in the proper operation of avalanche photodiodes. It was presented that HgCdTe (100) based N^+^-ν-p-P^+^ gain, *M* > 100 could be reached for reverse voltage > 5 V and excess noise factor *F*(*M*) assumes: 2.25 (active layer, *x_Cd_* = 0.22, *k* = 0.04, *M* = 10) for λ*_cut-off_* = 8 μm and *T* = 230 K. In addition the 4-TE cooled, 8 μm APDs performance was compared to the state-of-the-art for SWIR and MWIR APDs based mainly on III-V and HgCdTe materials (*T* = 77–300 K).

## 1. Introduction

The ability to detect light efficiently is increasingly important for applications in communications, security, medicine, and metrology. Conventional semiconductor photodiodes have limited sensitivity, especially in photon-deficiency and high-speed applications conditions such as free space optical communication and remote sensing. The most sensitive semiconductor light detector, that can detect the electromagnetic radiation of an extremely low intensity, is the avalanche photodiode (APD). This is possible due to the avalanche process of charge carriers. The incident photons generate primary electron-hole (e-h) pairs, then the generated carriers reaching the area of a strong electric field are accelerated attaining sufficient kinetic energy to generate more carriers due to the impact ionization mechanism. In this way, secondary carriers are created, which are also accelerated in the external field and create new carriers. The primary photocurrent is thus amplified from several to several million times depending on the applied voltage. Due to their internal multiplication gain, APDs overcome a fundamental limitation of traditional photodiodes, low sensitivity, thereby enabling them to be used to increase the signal-to-noise-ratio of a sensor. An APD can produce a better overall system signal-to-noise ratio than a p-n detector in cases where the APD internal gain boosts the signal level without dramatically affecting the overall system noise.

APDs with the highest gain can detect single photons. A high gain is their special advantage compared to classic photodiodes, however, they also have a number of disadvantages (see Table 1). It is generally preferred linear mode operation to detect weak signals. A variety of semiconductor material (to include III-V and II-VI) [1] and device design (PIN, SAM, SACM, submicron scaling of the multiplication region, impact ionization engineering (I^2^E)-staircase) [1,2,3] aspects have been taken into account to obtain high internal gain with low excess noise and high bandwidth to detect and amplify low optical signals.

These conditions are met by HgCdTe since the bandgap can be tuned to the particular requirement. HgCdTe APDs show a close to linear exponential gain up to *M* ~ 1000 with very low excess noise, close to *F*(*M*) ~ 1 [4,5,6,7,8,9,10,11] in short-wave (SWIR) mid-wave infrared (MWIR) to long-wave infrared (LWIR) covering niche strategic applications from 1.3 μm to 16 μm. Asymmetry between the effective mass of electron in the conduction band and heavy hole results in the unequal ionization coefficient for electron and hole in HgCdTe material. The band structure suggests that in a HgCdTe APD, electrons have a very small intervalley phonon scattering rate, which is the dominant scattering mechanism in most III-V semiconductors. In addition, the large effective mass ratio reaching ~30 indicates that there is a large difference in the phonon and alloy scattering rates of holes and electrons for transport in HgCdTe. This is supported by high electron mobility (two orders of magnitude higher than hole mobility) in HgCdTe materials. In addition, HgCdTe based APD operates at a low reverse bias to achieve large internal gain (*M*) with low excess noise factor [*F*(*M*)] rather than III-V material based APD. However, the highest performance is obtained at low cryogenic temperatures (*T* = 77 K) [4,5,6,7,8,9,10,11]. 

Recently the tendency related to reach the high operating temperature condition (HOT, temperature, *T* > 190 K) reached by 4, 3, 2-stage thermoelectric (TE) coolers has been observed in IR technology. That is directly related to the attempts to reduce the IR detector size, weight, and power dissipation (SWaP) conditions and possible field applications where cryogenic cooling is difficult to deploy. The room temperature avalanche photodiodes technology is well developed in SWIR while devices operating in MWIR and LWIR require cooling to suppress dark (mainly tunneling contribution) current due to the low bandgap. This is a major reason that MWIR to LWIR APDs operating at high temperatures are in their infancy but show great promise for applications across the IR spectrum, such as active imaging, laser Radar/Lidar, wavefront sensing, and photon counting.

In this paper, theoretical analysis has been performed to show the impact of the structural parameters of the multiplication region (its doping and thickness) on the dark currents and the gain in HgCdTe APD designed on an 8-µm cut-off (λ*_cut-off_*) wavelength and a temperature of 230 K. The calculation results for the N^+^-ν-p-P^+^ APD have also been pre-compared to the N^+^-p-P^+^-n^+^ HgCdTe (100) and HgCdTe (111)B photodiodes grown by metal-organic chemical vapor deposition (MOCVD) on GaAs substrates. Both crystallographic orientations differ in the level of residual doping, which also affects the minimum doping of the p-type. The minimal background concentration is *N_D_* ~ 5 × 10^14^ cm^−3^ and *N_D_* ~ 1 × 10^15^ cm^−3^ for the HgCdTe (100) and (111)B technology, respectively. This gives the possibility of controlled p-type doping of the absorber on the level of *N_A_* ~ 3 × 10^15^ cm^−3^ and *N_A_* ~ 2 × 10^16^ cm^−3^ for the HgCdTe (100) and (111)B technology, respectively. It was presented that HgCdTe (100) based N^+^-v-p-P^+^ gain, *M* > 100 could be reached for reverse voltage > 5 V and excess noise factor *F*(*M*) assumes: 2.25 (*x_Cd_* = 0.22, *k* = 0.04, *M* = 10) for λ*_cut-off_* = 8 μm and *T* = 230 K. In addition, the 4-TE cooled, 8 μm APDs performance was compared to the state-of-the-art for SWIR and MWIR APDs based mainly on III-V and HgCdTe materials (*T* = 77–300 K).

## 2. Performance Parameters of HgCdTe APDs

The avalanche effect occurs in the area of a strong electric field, which causes a multiplication of photogenerated charge carriers. A measurable indicator of this effect is the avalanche gain, which depends on many factors including the properties of the semiconductor material, as well as the design of the device. All these factors will be discussed below.

### 2.1. Impact Ionization

Impact ionization can proceed in two ways: ionization caused by both types of charge carriers, and ionization caused by one carrier type. If electrons and holes both ionize significantly, the hole to electron impact ionization ratio (*k* = *α_h_*/*α_e_*) is equal to unity (*k* ≈ 1). Although this type of process leads to the avalanche gain, it is undesirable to fabricate APDs for several reasons: the ionization time of two types of charges is long which increases the time constant of the device;it is random, and hence increases the excess noise of the device;it can be unstable, thereby causing avalanche breakdown.

Therefore, the latter case is more desirable to obtain a high avalanche gain with a low noise level. In some semiconductors, electrons ionize more efficiently than holes (for which *α_e_* > *α_h_*), while in others the opposite is true (where *α_h_* > *α_e_*). The ideal case of single-carrier multiplication is achieved when *k =* 0 or *k =* ∞. 

In Hg_1−*x*_Cd*_x_*Te, the hole to electron impact ionization ratio is dependent on the Cd composition (*x_Cd_*) [12,13]: electron injection for lower *x_Cd_*-values (*x_Cd_* < 0.5) or hole injection for 0.5 < *x_Cd_* < 0.7. The dependence of *k* on the molar composition in HgCdTe is described as follows: 

When Δ > *E*_g_ (Δ is the spin-orbit splitting energy) the electron ionization, which based on one-band transitions, is favorable. Due to conservation of energy and momentum, a threshold energy must be greater than the bandgap as the carrier also gain energy, lose energy, or exchange momentum.

When Δ ≤ *E*_g_ hole is dominant in the multiplication process. The energy required for the excitation of an electron from the valence band to the conduction is equal to the energy of spin-orbit splitting.

The ballistic transport model of the “lucky electron” discussed by Brennan [14], then modified for a Kane-type non-parabolic conduction band, gives the expression for the ionization coefficient [15].
(1)$$\alpha_{e}=\frac{\xi }{V_{ion}}\exp\left(\frac{-\sqrt{2m^{*}qV_{ion}}}{q\xi \tau }\right),$$
where *ξ* is the electric field, *V_ion_* is the voltage required for the impact ionization, *τ* is the relevant lifetime for momentum scattering of the electron.

### 2.2. Dark Current

Since the dark current will be multiplied in the device along with generated photocurrent, APDs must exhibit low dark currents. When the photodiode is reverse biased, a relatively small electric current flows through the device. Physically, the saturation current is due to the thermal generation of e-h pairs within the diffusion and depletion region of the photodiode. 

The diffusion current arises from the thermal generation of carriers in the un-depleted absorber. Taking into account the radiative, Auger and Shockley-Read-Hall (SRH) mechanisms, the diffusion current density can be expressed as
(2)$$J_{dif}=\frac{qn_{i}^{2}t_{dif}}{N_{maj}}\left(\frac{1}{\varphi \tau_{R}}+\frac{1}{\tau_{A}}+\frac{1}{\tau_{SRH}}\right),$$
where *n_i_* is the intrinsic carrier concentration, *t_dif_* is the diffusion region thickness, *N_maj_* is the majority carrier concentration. *τ_R_* is the lifetime due to radiative recombination, *φ* is the product of the photon recycling factor for radiative recombination, *τ_A_* is the lifetime due to Auger recombination, and *τ_SRH_* is the lifetime for recombination through SRH centers in the bandgap. 

The second source of dark current derives from the portion of the semiconductor that becomes depleted. The depletion current density can be estimated by the following expression
(3)$$J_{GR}=\frac{qn_{i}W}{\tau_{SRH}},$$
where *W* is the width of depletion region.

This current source depends on the density of defects located in the space-charge region. Thus, the SRH carrier lifetime is a sensitive indicator of material quality, being strongly affected by defects’ concentration acting as recombination centers. Since it is not a fundamental limitation of the photodiode, it might be overcome by a careful optimization of material technology.

As-grown HgCdTe is often limited by the mercury vacancies [16,17], whose concentration can be effectively reduced by the thermal annealing during the controlled cooling after the growth. However, the reverse-bias characteristics of HgCdTe diodes are strongly dependent on dislocations intercepting the junction [18]. The high dislocation density in HgCdTe alloys grown on GaAs is due to the large lattice mismatch (14%) between CdTe and GaAs. The physics of such crystallographic defect in HgCdTe is complex and differs in HgCdTe (111)B and HgCdTe (100) layers. Misfit dislocations in the HgCdTe (111)B layers are concentrated near the interfaces due to the slip planes parallel to these interfaces and their density remains almost constant throughout the epi-layer thickness. In the HgCdTe (100) layers, dislocations propagate monotonically to the growth surface and their density decreases gradually from the layer interface as dislocation lines approach each other. Such different behavior of the type of dislocations is also visible in the device current, especially under strong reverse bias.

Under moderate reverse bias, additional undesirable tunneling currents [19], due to direct band-to-band transitions and/or transitions via trap states within the bandgap, may appear. If the dark current contribution appears into the photocurrent, it is difficult to distinguish the avalanche effect from the tunneling one [20,21]

Band-to-band tunneling (BtBT) for a parabolic barrier in a uniform electric field is given by
(4)$$J_{BtBT}=\frac{q^{3}{\left(2m^{*}\right)}^{1/2}\xi V}{4\pi^{3}\hbar^{2}E_{g}^{1/2}}\exp\left(\frac{-\pi m^{*1/2}E_{g}^{3/2}}{2\sqrt{2}q\xi \hbar }\right),$$
where *m**^∗^* is the electron effective mass, *ħ* is the reduced Planck constant, *E_g_* is the semiconductor energy gap.

Trap-assisted tunneling (TAT) through bandgap states is given by
(5)$$J_{TAT}=\frac{q^{2}m^{*}V\kappa^{2}N_{T}}{8\pi \hbar^{3}\left(E_{g}-E_{T}\right)}\exp\left(\frac{-m^{*1/2}E_{g}^{3/2}G\left(a\right)}{2\sqrt{2}q\xi \hbar }\right),$$
(6)$$\kappa =\frac{2\sqrt{2\pi }\hbar^{2}}{m_{0}}{\left(\frac{m_{0}}{\hbar^{2}}\right)}^{1/4}\frac{E_{g}}{{\left(0.5E_{g}\right)}^{3/4}},$$
where $\kappa^{2}$ is a matrix element associated with the defect potential [22], *N_T_* is the density of centers at *E_T_*, and *G*(*a*) is the geometrical factor in the exponent.

The total bulk dark current-voltage equation is expressed as [15]
(7)$$J_{DB}=J_{dif}\left[\exp\left(\frac{qV}{kT}\right)-1\right]+J_{GR}\left[\exp\left(\frac{qV}{2kT}\right)-1\right]-J_{BtBT}-J_{TAT}.$$

In strong reverse bias, such that V>>kT/q (*V* is the bias voltage), we have
(8)$$J_{DB}=-\left[J_{dif}+J_{GR}+J_{BtBT}+J_{TAT}\right].$$

All dark current components of a typical HgCdTe photodiode, which consists of a p-type absorber, ν-type depletion region, and n^+^-contact region, are included in the schematic photodiode band diagram illustrated in Figure 1. It should be noted that this is the dark current flowing through the photodiode before the gain. In the APD, not only photocurrent, but also dark bulk current is multiplied by the gain. Therefore, for the APD efficiency to be as high as possible, each dark current component should be reduced. It is of interest to compare the diffusion current from the field-free volume given by Equation (2) with the depletion current of Equation (3), as well as all tunneling currents given by Equations (4) and (5). 

### 2.3. Avalanche Gain

Since in SWIR to LWIR HgCdTe *k* values are of 0.1 or less, only electrons are involved in the multiplication process. The avalanche gain, *M*, for an electron-APD (e-APD) is given by
(9)$$M=\exp\left(\alpha_{e}W\right).$$

The gain of APD can be achieved in the entire depletion region *W* assuming a constant distribution of the electric field *ξ* and an applied reverse bias *V* over the junction width, giving *ξ* = *V*/*W*. The gain in a uniform electric field is given by
(10)$$M=\exp\left[\frac{V}{V_{ion}}\exp\left(\frac{-2W\sqrt{2m^{*}qV_{ion}}}{qV\tau }\right)\right].$$

It should be noted that a constant distribution of the electric field could only be proposed in the ultra-low concentration in the ν-multiplication region.

### 2.4. Excess Noise

The excess noise in APDs arises from the random nature of the impact ionization process and limits the device’s detectivity. In the multiplication region, the bulk leakage dark current density *J_DB_*, is multiplied by the gain, *M*, of the APD. The total leakage dark current density *J_D_* is therefore equal to
(11)$$J_{D}=J_{DS}+J_{DB}M,$$
where *J_DS_* is the surface leakage current component.

The total spectral noise current for an APD in dark conditions is thus given by:(12)$$I_{S}={\left[2qA\left(J+J_{DB}M^{2}F\left(M\right)\right)B\right]}^{1/2},$$
where *A* is the detector area, *F*(*M*) is the excess noise factor, *B* is the system bandwidth.

In illuminated conditions, the detector transitions to the photon shot noise limited regime. The total noise from the APD is then equal to
(13)$$I_{S\left(TOTAL\right)}={\left\{2qA\left[J_{DS}+\left(J_{DB}+J_{UPh}\right)M^{2}F\left(M\right)\right]B\right\}}^{1/2},$$
where *F*(*M*) is the *J_UPh_* is the density of unity gain photocurrent (photocurrent at *M* = 1).

The excess noise is characterized by the excess noise factor *F*(*M*), which depends on *k* and *M*. When electrons initiate the multiplication, *F*(*M*) can be calculated according to the McIntyre noise theory [23]
(14)$$F\left(M\right)=M\left[1-\left(1-k\right){\left(\frac{M-1}{M}\right)}^{2}\right].$$

To minimize the noise current, a large asymmetry between the impact ionization ratio of holes and electrons is advantageous. Hg_1−x_Cd_x_Te band structure with molar composition *x_Cd_* < 0.5 gives *k* values of 0.1 or less-a very favorable hole-to-electron ratio under avalanche conditions, resulting in increased gain without generating excess noise. Those properties give HgCdTe a figure of merit for the design of e-APDs operating in the MWIR and LWIR ranges.

## 3. Experimental Studies of Dark Current

The multiplication effect should take place in the region with low doping concentration to provide high resistance, and consequently sufficient strength of the electric field distribution, as well as to avoid the tunneling current contribution at higher bias. Moreover, in order to reduce mainly tunnel currents via trap states in the bandgap, a high quality of the epitaxial layer with a low number of defects is required. The level of residual doping and the number of defects depends on the growth technology. 

In order to determine the level of technology, the analysis of dark currents was performed in MOCVD grown HgCdTe (100) oriented and HgCdTe (111)B photodiodes. HgCdTe heterostructures were grown in a horizontal Aixtron AIX-200 MOCVD system on 2-inch, epiready, semi-insulating (100) GaAs substrates. CdTe buffer layers were deposited prior to HgCdTe. Recent paper shows [24] that the residual donor concentration of HgCdTe (100) epilayers is in the mid-range of *N_D_* = 10^14^ cm^−3^, and is significantly lower compared to (111)B ones. It is primarily important not only for the low doping required for the multiplication layer but also for ensuring an appropriate level of p-type doping of an active layer. 

Analyzed in the paper photodiodes have a backside-illuminated in a mesa-type N^+^-p-P^+^-n^+^ structure (Figure 2). A wider-bandgap N^+^-type layer with a doping level of *N_D_* = 2 × 10^17^ cm^−3^ was formed into a buffer layer. It consists of the bottom contact but also acts as an optical window for IR radiation. Photon absorption occurs in the p-type doped absorber with arsenic (As) at the level of *N_A_* = 3.5 × 10^15^ cm^−3^ and *N_A_* = 2.5 × 10^16^ cm^−3^ in HgCdTe (100) and HgCdTe (111)B photodiode, respectively. The Cd molar composition in the active area was chosen to obtain a *λ_cut-off_* ~ 8 μm at 230 K (Figure 2). The absorber is sandwiched between highly doped N^+^ contact and P^+^-type wider-bandgap barrier layer with an acceptor doping level of *N_A_* = 5 × 10^17^ cm^−3^. There is also a thin n^+^-type layer on the top to improve the ohmic contact. All interfaces contain the *x*_Cd_-graded regions created by interdiffusion processes during HgCdTe growth to prevent energy band discontinuities. 

The detector architecture with wider-bandgap contacts (P^+^ and N^+^ volumes) is used to obtain the exclusion and extraction of charge carriers from the absorber. Under a moderate reverse bias, the minority carriers (electron) thermally generated at the absorber are fully extracted, while the majority carrier (hole) concentration is reduced to the doping level which causes a suppression of the Auger g-r process [25,26].

The current peak responsivity (*R_i_*) at λ = 5.9 μm is two times higher for (100) orientation reaching ~3.1 A/W and ~1.7 A/W for (111)B. The inset presents peak responsivity distribution versus voltage for both analyzed N^+^-p-P^+^-n^+^ structures (*T* = 230 K). The current responsivity of the HgCdTe (111)B photodiode decreases with increasing reverse voltage (see the inset of Figure 3). The decrease in responsivity is related to the increase in tunnel currents, which is not visible in the HgCdTe (100) one. 

Figure 4 shows the dark current density as a function of bias for HgCdTe photodiodes measured at 230 K. The relative magnitudes of the diffusion, depletion, and tunneling current components were simulated with assumed parameters presented in Table 2 and are also shown in Figure 4. For the photodiode with non-injecting minority carrier contacts-these can be provided by an N^+^-p and p-P^+^ heterojunctions-the dark current stems only from field-free volume of the active layer and the depleted part of the absorber. The maximum current density reaches ~3 A/cm^2^ and the Auger suppression is observed for *V* > 0.05 V for (100) orientation while (111)B reaches ~20 A/cm^2^ and negative differential resistance region is not reached. In addition, for the HgCdTe (100) oriented photodiode, the diffusion current given by Equation (2) reads ~2.12 A/cm^2^ for an assumed doping concentration of *N_A_* = 3 × 10^15^ cm^−3^ and the SRH carrier lifetime component of *τ_SRH_* = 135 ns. For the HgCdTe (111)B photodiode, the diffusion current is higher and reaches 4.3 A/cm^2^ for an assumed doping concentration of *N_A_* = 2 × 10^16^ cm^−3^ and *τ_SRH_* = 66 ns. In the HgCdTe (100) photodiode, despite the higher SRH carrier lifetime, the depletion current given by Equation (3) is higher than in HgCdTe (111)B one due to the wider space charge region (*W*).

HgCdTe (111)B photodiode shows a dominant TAT mechanism, given by Equation (5), for higher voltages. The component of tunneling current via a direct band-to-band mechanism, as given by Equation (4), is also observed for the chosen doping concentration of *N_A_* = 2 × 10^16^ cm^−3^. The contribution of the tunneling currents to the saturation current from the neutral region affects the performance of the device at high reverse bias. 

Increased TAT in (111)B photodiode compared to (100) one is believed to be associated with trap density of *N_T_* = 1 × 10^15^ cm^−3^ in the semiconductor bandgap immediately surrounding the dislocation with a capture coefficient of 3 × 10^–8^ cm^3^/s and a resulting lifetime of *τ_SRH_* = 66 ns. 

Compared to the devices analyzed above, the APD has a separate multiplication region-ν-multiplication region incorporated between the p-type absorber and N^+^-type contact layer: N^+^-ν-p-P^+^ design (SAM-separated absorption and multiplication). Once the electrons reach the ν-multiplication region from the p-absorber, the avalanche multiplication process takes place. In order to suppress the dark current influence on the photocurrent, especially related to tunnel currents, an appropriate doping in the ν-multiplication region is required.

Figure 5 shows the dependence of the dark current as a function of reverse bias for various doping concentrations of ν-region. The N^+^-p-P^+^-n^+^ HgCdTe trap material parameters were implemented into the N^+^-ν-p-P^+^ SAM structure. The theoretically simulated data shows the difference in dark current with TAT and without TAT contribution. The width of the avalanche multiplication region was assumed to be 2 µm. In the first attempt, calculations were made with the assumption of both BtBT and TAT mechanisms Figure 5a. The density of mid-bandgap states of *N_T_* = 1 × 10^15^ cm^−3^ with the corresponding trap capture coefficient of 3 × 10^–8^ cm^3^ s^−1^ and the ionization energy of 0.85 × *E_g_* were assumed. Figure 4a shows the boundary between the dominance of the TAT and BtBT mechanisms. 

With lower doping of the ν-multiplication region (*N_D_* < 10^15^ cm^−3^), TAT is the dominant tunneling mechanism in the entire range of the analyzed reverse voltage. For this reason, the current is considerably larger than purely thermal current for these doping levels. Interband tunneling occurs at a higher dopant concentration. The calculations show that not only the reduction of the dopant concentration in the ν-multiplication region is necessary for the correct operation of the APD, it is also necessary to have a good quality material with a low trap concentration. Figure 5b shows calculations made with only the BtBT mechanism-that is, with a reduced concentration of mid-bandgap states. A low doping concentration (*N_D_* < 10^15^ cm^−3^) in the ν-multiplication region is required to reduce tunneling currents.

## 4. Gain Calculation

To achieve a uniform electric field in the multiplication region, the doping-thickness product should be made adequately low to fully deplete the ν-multiplication region under the appropriate reverse-bias. Figure 6 shows the dependence of the depletion width on bias (*V*) and doping (*N_D_*), calculated as $W={\left[2\varepsilon \varepsilon_{0}\left(E_{g\nu }+V\right)/qN_{D}\right]}^{1/2}$ where *E_gν_* is the multiplication region energy gap. For the range of doping achievable at HgCdTe (100) technology (*N_D_* ~ 5 × 10^14^ cm^−3^) [21], a 2-μm-thick ν-multiplication region can be fully depleted with the reverse bias above 0.9 V. By reducing the background concentration to 2 × 10^14^ cm^−3^, a 2-μm-thick ν-multiplication region is fully depleted at a reverse bias of 0.25 V. For HgCdTe (111)B material with a background concentration *N_D_* > 2 × 10^15^ cm^−3^, a complete depletion of the 2-μm layer is reached above the voltage of −2 V.

In a uniform electric field, the gain determined by Equation (9) is seen to be dependent only on the ionization voltage *V_ion_*, which is related to the electron ionization energy *E_ion_* by the relation *V_ion_* = 2*E_ion_*. A value that is in agreement with the experimental results for MWIR and LWIR HgCdTe e-APDs is *V_ion_* = 5*E_g_* [14]. The modeled gain-voltage characteristics for bandgap value of the ν-multiplication region of *E_g_* = 0.16 eV and different depletion widths (*W* = 2 and 4 μm) are shown in Figure 7. 

A momentum scattering lifetime of 10^−12^ s and the voltage required for the impact ionization *V_ion_* = 5*E_g_* have been assumed. For the molar composition *x_Cd_* = 0.22 in the ν-multiplication region this gives the ionization voltage value of 0.8 V. This is close to the value at which full-depletion of the 2-μm-thick multiplication region doped at level 5 × 10^14^ cm^−3^ is obtained. The threshold voltage, that is the applied reverse voltage at which the gain, *M* = 2 is 1.4 V and 2 V for the 2-μm and 4 μm-thick ν-multiplication regions, respectively. This gives the corresponding electric field values of 7 × 10^3^ V/cm and 5 × 10^3^ V/cm, respectively.

## 5. Comparison of *λ* ~ 8 μm (230 K) HgCdTe versus SWIR and MWIR APDs

Table 3 presents SWIR [27,28,29,30,31,32,33,34,35,36,37,38,39,40,41,42,43,44], MWIR [45,46,47,48,49,50] (PIN, SAM and SACM) state-of-art and LWIR, λ ~ 8 μm (230 K) HgCdTe SAM APDs based devices to include maximum gain (*M*), impact ionization ratio (*k*), excess noise factor *F*(*M*) and dark current mostly at *M* = 10. As presented, the majority of the published papers corresponds to the *T* = 300 K and SWIR range. The cryogenically cooled (*T* = 77 K) HgCdTe PIN are well developed reaching *M* = 5300, *k* < 0.001 and *F*(*M*) = 1–1.3 depending on the growth technology [47,49]. An increase in temperature by 50 K decreases *M* to the level of 120 [47]. APDs based on T2SLs InAs/GaSb are theoretically predicted to be able to achieve higher gains and higher breakdown voltages with lower noise than HgCdTe APDs and hence outperform the HgCdTe APDs in high gain applications. The basis of the superior high gain performance of the T2SLs APDs rests on the ability to engineer the band structure so as to maintain *k* different from 1 to much higher fields than HgCdTe alloys. This flexibility to engineer the band structure of T2SLs is a fundamental property that cannot be achieved in bulk semiconductors. This allows, in principle, to have either electron- or hole-dominated avalanche multiplication by engineering the higher lying energy bands. As of now, in the MWIR range, T2SLs InAs/GaSb PIN APDs reach the comparable *M*, *k*, and *F*(*M*) values compared to the HgCdTe [47,48]. The promising results were presented by Li et al. reporting on MWIR λ = 5 μm, *T* = 200 K (4-stage TE cooling) *k* = 0.097, *M* = 29 and *F*(*M*) = 4.98 AlAsSb/GaSb SL SAM device [46]. The MWIR λ = 4.6 μm, *T* = 150 K InAs/InSb SL PIN device was presented by Dehzangi et al. reporting on *k* = 0.27, *M* = 6 and *F*(*M*) = 2.95 [45]. The impact ionization coefficient for that structure is acceptable, but it is still much greater than the reported cases for HgCdTe and T2SLs InAs/GaSb APDs (*k* < 0.001) [48].

Figure 8 presents the comparison between SWIR (λ = 1.5 μm, *T* = 300 K) state-of-art of excess noise *F*(*M*) versus gain (*M*) based on III-V materials (AlAsSb, AlInAsSb, AlGaSb, InAsAs, InAlAs, digital alloy-DA InAlAs) and MWIR (*λ* = 4.6–5 μm, *T* = 120–200 K) HgCdTe, T2SLs InAs/GaSb, SL InAs/InSb, SL AlAsSb/GaSb and LWIR HgCdTe (λ = 8 μm, *T* = 230 K) HgCdTe. 

Reported for SWIR experimentally measured ionization coefficient ratio *k* = α_h_/α_e_ values follows McIntyre’s theory and assumes 0.005 < *k* < 0.2 suggesting on single carrier ionization process and allows to reach *F*(*M*) within the range 1.8 *< F*(*M*) < 3.5. 

In terms of the HgCdTe and T2SLs InAs/GaSb the value of the ionization coefficient ratio *k* is close to zero (*k* < 0.001). The McIntyre classical model predicts a noise factor of 2 for *k*~0 and for *x_Cd_* = 0.22–0.3 ionization coefficient ratio, *k* = 0.04. The discrepancy in the measured and theoretically estimated *F*(*M*) is related to the “*dead space*” effect where the carriers must travel a minimum distance before they can gain enough energy to impact ionize, whereas McIntyre theory assumes the impact ionization to be dependent only on the electric field of the carrier’s current position. The AlAsSb/GaSb SL [46] SAM carrier ionization ratio assumes *k* ~ 0.097 at 200 K, which is smaller than the value of *k* ~ 0.27 achieved for InAs/InSb SL [45] PIN. 

## 6. Conclusions

Assuming HgCdTe SAM structure the multiplication effect should take place in the region with low doping concentration to provide high resistance, and consequently sufficient strength of the electric field distribution, as well as to avoid the tunneling current contribution at higher bias. Moreover, in order to reduce mainly tunneling currents via trap states in the bandgap, a high quality epitaxial layer with a low number of defects is required. The level of residual doping and the number of defects depend on the growth technology. 

Experimental results and calculations show that the HgCdTe (100) layers exhibit better crystalline quality than HgCdTe (111)B, low dislocation density and reduction of residual defects, which contribute to a background doping in the mid of 10^14^ cm^–3^. Calculations fitting to the experimental dark currents of the N^+^-p-P^+^-n^+^ photodiodes operating at 230 K allowed us to determine the parameters of the material. An assumed SRH carrier lifetime of 66 ns for HgCdTe (111)B photodiode would suggest a measurable density of mid-bandgap states of *N_T_* = 1 × 10^15^ cm^−3^. This causes a dominant TAT mechanism for higher reverse voltages. The SRH lifetime of 135 ns for HgCdTe (100) photodiode gives the density of mid-bandgap states of *N_T_* = 2.5 × 10^14^ cm^−3^ with visibly limited tunneling currents. Moreover, a voltage required for the impact ionization in HgCdTe material with the energy gap of 0.16 eV is 0.8 V. 

In order to take full advantage of the ionization effect, it is required that the region in which it takes place at this voltage was fully-depleted to ensure an appropriate high and uniform electric field. For the range of doping achievable at HgCdTe (100) technology (*N_D_* ~ 5 × 10^14^ cm^−3^), a 2-μm-thick ν-multiplication region can be fully depleted with the reverse bias above 0.9 V. For HgCdTe (111)B material with a background concentration *N_D_* > 1 × 10^15^ cm^−3^, a complete depletion of the 2-um layer is obtained above the voltage of −2 V. It was presented that HgCdTe (100) based N^+^-ν-p-P^+^ gain, *M* > 100 could be reached for reverse voltage > 5 V and corresponding excess noise *F*(*M*) assumes: 2.25 for λ*_cut-off_* = 8 μm and *T* = 230 K. In addition, the 4-TE cooled, 8 μm APDs performance was compared to the state-of-the-art for SWIR and MWIR APDs based mainly on III-V materials (*T* = 300 K).

Taking into account the HgCdTe material parameters and the properties resulting from the MOCVD technique, the following APD architecture was proposed:wider-bandgap N^+^-type bottom contact layer with a doping level of *N_D_* ~ 2 × 10^17^ cm^−3^. Cd molar composition is much greater (*x_Cd_* ~ 0.4) than that of the absorber so that it is also an optical window for the IR radiation. Layer thickness sufficiently large (~9 μm) to perform “mesa” structure etching.non intentionally doped ν-multiplication region (*N_D_* ~ 5 × 10^14^ cm^−3^) with Cd molar composition slightly greater (*x_Cd_* ~ 0.22) than that of the absorber. Layer thickness in the order of 2–2.5 µm to achieve complete depletion after crossing the ionization voltage.p-type absorber doped at the level of *N_A_* ~ 3 × 10^15^ cm^−3^ with Cd molar composition of *x_Cd_* ~ 0.21 to obtain a λ*_cut-off_* ~ 8 μm at 230 K. Thickness of about 5 μm optimized for the best compromise between requirements of efficient collection of IR radiation and low thermal generation;wider-bandgap P^+^-type barrier layer with a doping level of *N_D_* ~ 5 × 10^17^ cm^−3^. Cd molar composition of *x_Cd_* ~ 0.31 with a programed dopant and compositional gradient at the absorber side. Layer thickness not less than 0.6 µm so that it does not diffuse during the growth;narrow bandgap (*x_Cd_* ~ 0.13), heavily doped (*N_D_* ~ 3 × 10^17^ cm^−3^) n^+^-type cap contact layer. Such design should create a tunneling junction between the absorber and cap contact layer.

The N^+^-ν-p-P^+^-n^+^ structure is shown in Figure 9. The main layers are interfaced with thin graded gap and doping level transition layers that are formed by diffusion processes during growth of the nominally layered structure or introduced with programmed growth.

## Figures and Tables

**Figure 1 sensors-22-00924-f001:**
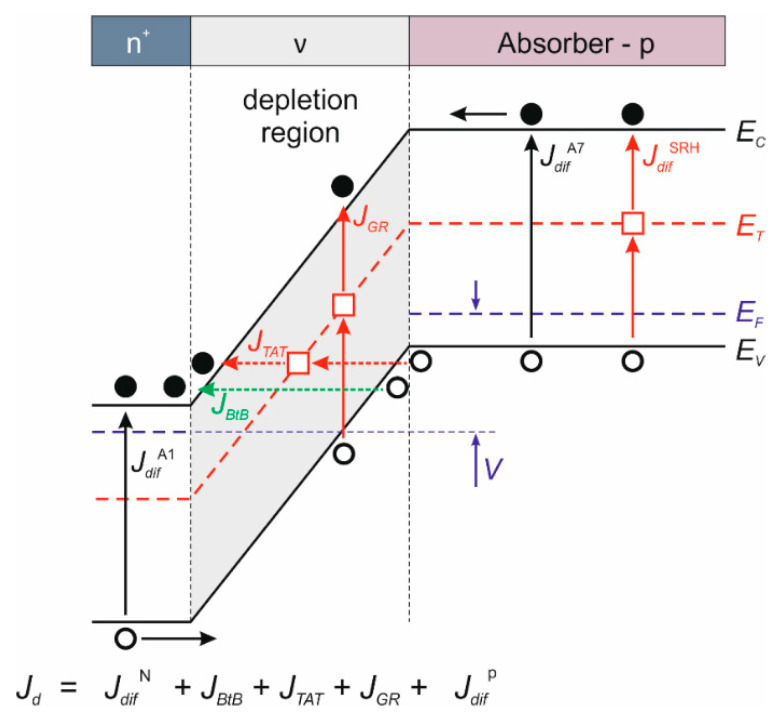
Schematic band diagram with relevant dark current components.

**Figure 2 sensors-22-00924-f002:**
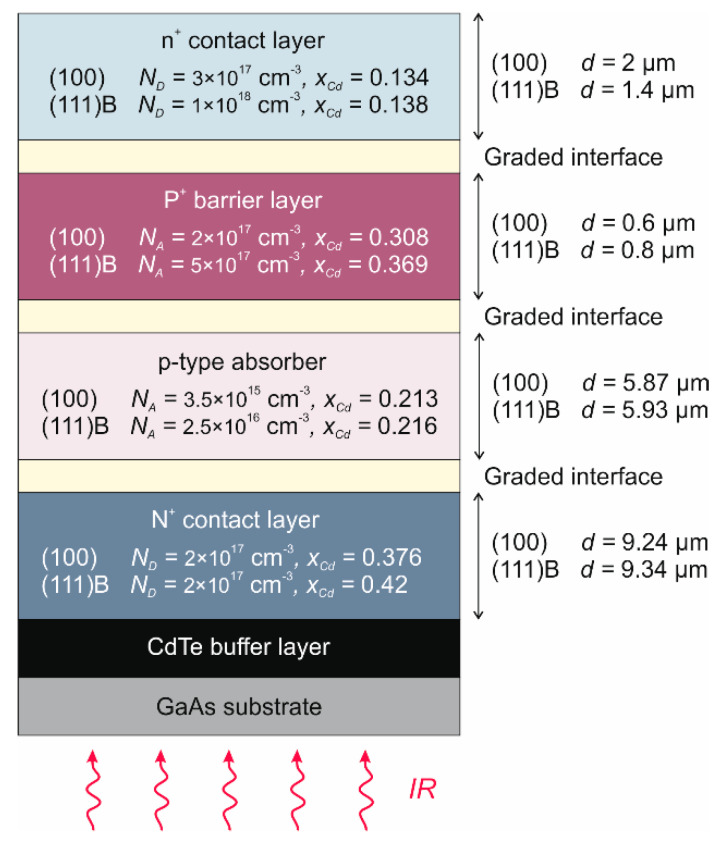
Layer structure of N^+^-p-P^+^-n^+^ HgCdTe photodiode.

**Figure 3 sensors-22-00924-f003:**
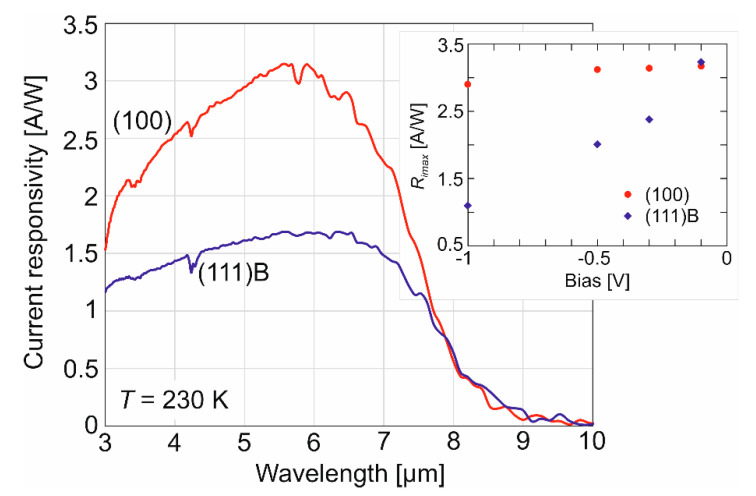
Spectral current responsivity of N^+^-p-P^+^-n^+^ HgCdTe photodiodes measured at 230 K. Inset: Maximum current responsivity versus bias.

**Figure 4 sensors-22-00924-f004:**
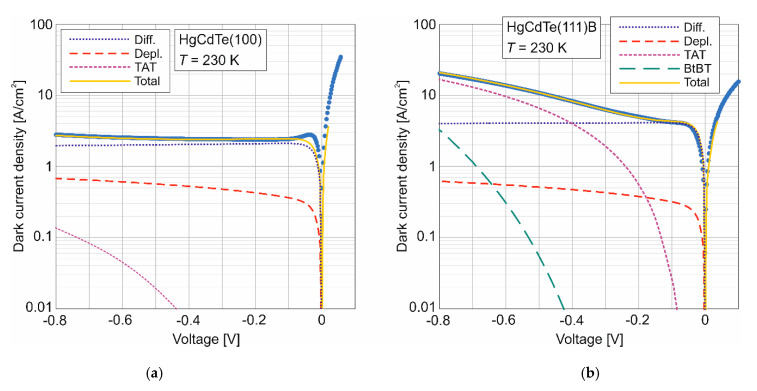
Measured at 230 K and calculated dark current density (with shown component contribution) as a function of bias for N^+^-p-P^+^-n^+^ HgCdTe (100) (**a**) and HgCdTe (111)B (**b**) photodiode.

**Figure 5 sensors-22-00924-f005:**
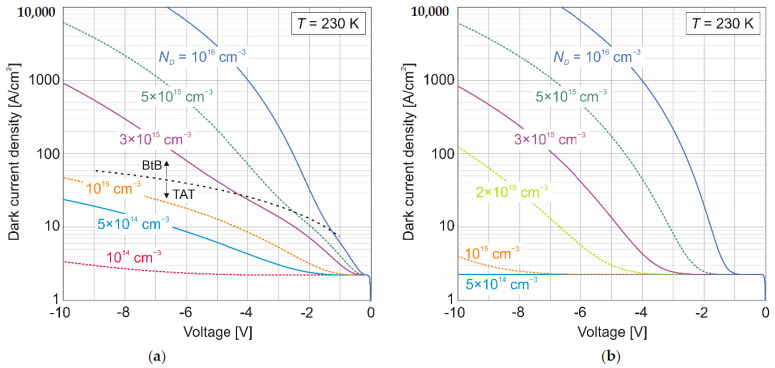
Dark current density versus bias for N^+^-ν-p-P^+^ HgCdTe APD calculated for various doping concentration of ν-multiplication region: (**a**) dark current with TAT and (**b**) without TAT.

**Figure 6 sensors-22-00924-f006:**
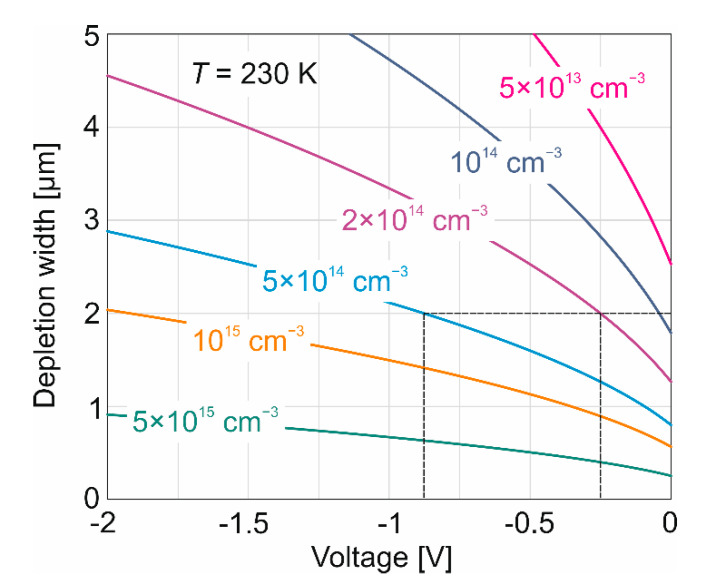
Calculated HgCdTe depletion width versus reverse voltage and doping concentration.

**Figure 7 sensors-22-00924-f007:**
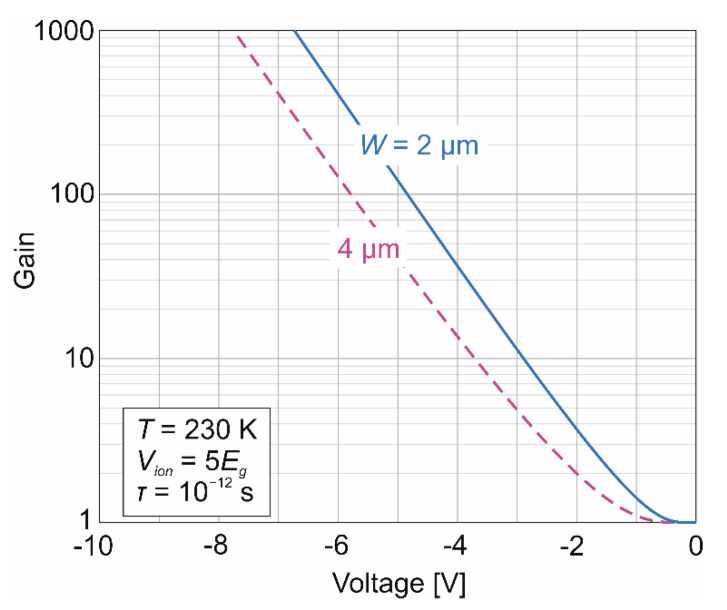
Calculated gain versus applied voltage for N^+^-ν-p-P^+^ HgCdTe APD. Calculations have been done for the 2-μm and 4 μm-thick ν-multiplication regions doped at level of 5 × 10^14^ cm^−3^.

**Figure 8 sensors-22-00924-f008:**
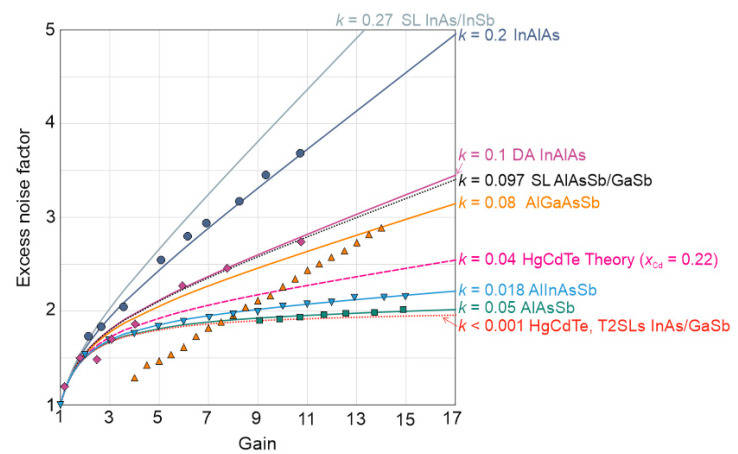
Excess noise factor *F*(*M*) versus gain (*M*) for III-V selected materials and HgCdTe SWIR, MWIR and LWIR ranges. Solid black lines are theoretical excess noise values predicted using McIntyre’s local field model. Data points are the measured excess noise [44]. Reprinted with permission from Ref. [44]. Copyright 2021. AIP Applied Physics Letters.

**Figure 9 sensors-22-00924-f009:**
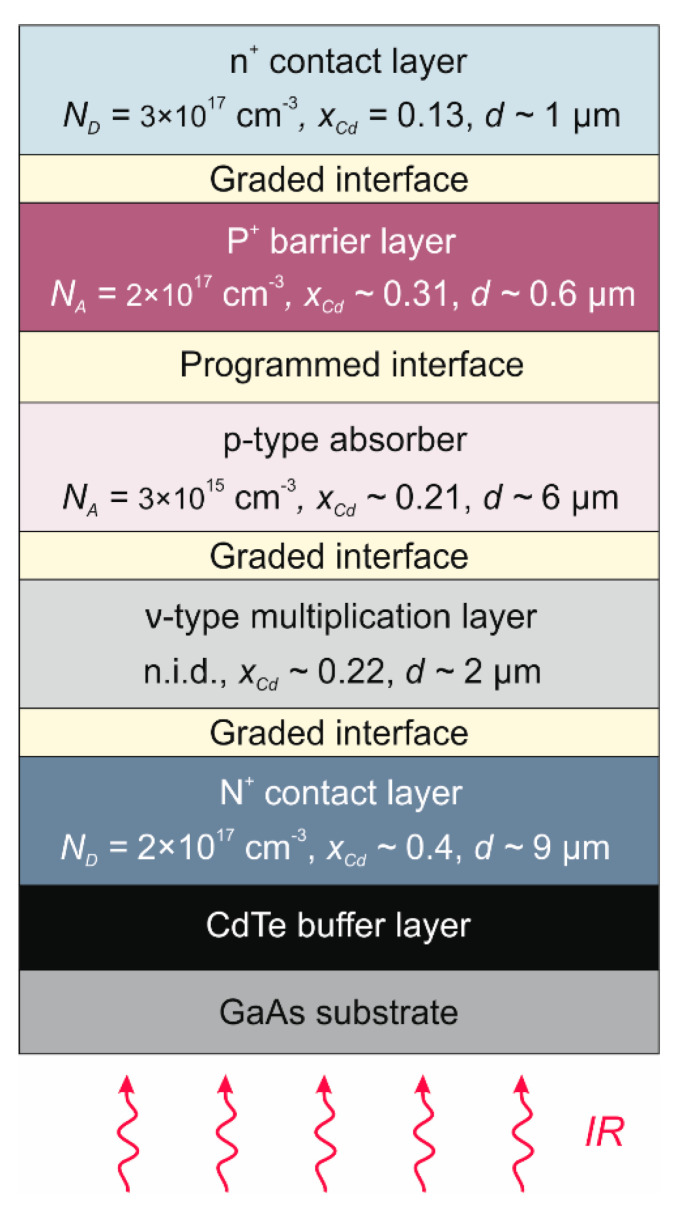
General idea of LWIR HOT HgCdTe APD with graded interfaces.

**Table 1 sensors-22-00924-t001:** Advantages and disadvantages of an APD.

Advantages	Disadvantages
high level of sensitivity as a result of avalanche gain.	high operating voltage; high excess noise level; non-linear output due to the avalanche process; strong dependence of sensitivity on bias voltage and temperature.

**Table 2 sensors-22-00924-t002:** Simulation parameters of N^+^-p-P^+^-n^+^ HgCdTe photodiodes.

Parameters	(100)	(111)B
Bandgap energy, *E_g_* [eV]	$E_{g}\left(x,T\right)=-0.302+1.93x-0.81x^{2}+0.832x^{3}+5.35\times 10^{-4}\left(1-2x\right)T$	
Intrinsic concentration, *n_i_* [cm^−3^]	$\begin{array}{c} n_{i}=\left(5.585-3.82x+0.001753T+0.001364xT\right)\\ \times 10^{14}E_{g}^{3/4}T^{3/2}\exp\left(-\frac{E_{g}}{2k_{B}T}\right) \end{array}$	
$\mathrm{Static}\;\mathrm{dielectric}\;\mathrm{constant},\;\varepsilon_{S}$	$\varepsilon_{s}=20.5-15.5x+5.7x^{2}$	
$\mathrm{High}-\mathrm{frequency}\;\mathrm{dielectric}\;\mathrm{constant},\;\varepsilon_{\infty }$	$\varepsilon_{\infty }=15.2-15.6x+8.2x^{2}$	
Cd composition in absorption region, *x**_Abs_*	0.213	0.216
Doping in absorption region, *N**_A_* [cm^−3^]	3 × 10^15^	2 × 10^16^
Absorber thickness, *d**_Abs_* [μm]	5.1	5.9
Trap concentration, *N**_T_* [cm^−3^]	2.5 × 10^14^	1 × 10^15^
Trap ionisation energy, *E_T_*	0.85 × *E_g_*	0.85 × *E_g_*
Trap capture coefficient, *γ* = *σv_yh_* [cm^3^ s^−1^]	3 × 10^–8^	1.5 × 10^–8^
SRH carrier lifetime, *τ**_SRH_* [ns]	135	66
Electron effective mass$,\;m_{e}^{*}/m_{0}$	0.071 × *E_g_*	0.071 × *E_g_*
$\mathrm{Hole}\;\mathrm{effective}\;\mathrm{mass},\;m_{hh}^{*}/m_{0}$	0.65	0.65
Overlap matrix *F*_1_*F*_2_	0.15	0.2
Operating temperature, *T* [K]	230	230

*σ* is the capture cross-section for carrier, and *v_th_* is the thermal velocity.

**Table 3 sensors-22-00924-t003:** Status of the λ ~ 8 μm (230 K) HgCdTe SAM APDs versus 77–300 K III-V and HgCdTe avalanche-based devices (PIN and SAM, SACM).

IR Range	Material	Maximum *M*	*k*	*F*(*M*) @ *M* = 10	*J_DARK_*(A/cm^2^)@*M* = 10
SWIR λ = 1.5 μm, *T* = 125 K, *T* = 300 K	InGaAs [27,28,29]	14	03–0.5	4.33–5.93	0.94 × 10^−3^
	InGaAs/InP [30,31,32,33] SACM	200	0.4–0.5	5.14–5.95	5.1 × 10^−6^–8 × 10^−4^
	InGaAs/InAlAs [34,35,36] SACM	200	0.15	3.11–3.52	3.2 × 10^−4^–2.1 × 10^−3^
	AlGaAsSb [37] PIN	42	0–0.01	1.9–1.98	1.5 × 10^−4^
	DA InAlAs [38] PIN	24	0.01	<2	1.1 × 10^−2^
	AlAsSb [39] PIN	37	0.005	1.96	5.7 × 10^−2^
	AlGaInAs [40] PIN	25	0–0.22	<2	0.26
	Ge/Si [41] SACM	24	0.02	2	0.33
	HgCdTe [42] PIN	>100	0	1	>3 × 10^−4^ (125 K)
	AlInAsSb/AlInAsSb [43] SACM	50	0.01	2	4.6 × 10^−3^
	InGaAs/AlInAsSb [44] SACM	20	0.018	1.99	5.5 × 10^−5^
MWIR λ = 4.6 μm, *T* = 150 K	InAs/InSb SL [45] PIN	6 (6.5 V, 150 K)	0.27 ^Exp^	2.95	5 (*M* = 6, 150 K)
MWIR λ = 5 μm, *T* = 200 K	AlAsSb/GaSb SL [46] SAM	29 (14.7 V, 200 K) 121 (150 K)	0.097 ^Exp^ (200 K)	4.58	0.15
MWIR λ = 4.9 μm, *T* = 77 K	HgCdTe on Si substrate [47] PIN	1250 (10 V, 77 K) 200 (120 K)	<0.001 ^Exp^	1–1.2	0.0625 (*M* = 250)
MWIR λ = 4.9 μm, *T* = 77 K	T2SLs InAs/GaSb on GaSb substrate [48] PIN	1800 (20 V, 77 K) 200 (120 K)	<0.001 ^Exp^	1–1.2	6.25 (*M* = 250)
MWIR λ ~ 5 μm, *T* = 77 K	HgCdTe CdZnTe substrate [49] PIN	5300 (12.5 V)	<0.001 ^Exp^	1–1.3	2.7 × 10^−7^ (*M* = 5300)
MWIR, λ ~ 5 μm, *T* = 80 K	HgCdTe [50] PIN Guard ring	400 (8 V)	0.04 ^Theory^	2.24	111 (8 V, *M* = 400)
LWIR λ = 8 μm, *T* = 230 K	HgCdTe SAM	>100 (>5)	0.04 ^Theory^	2.25	11 (5 V)

^Exp^ means the experimental data and ^Theory^ means the theoretical data.

## Data Availability

Not applicable.

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
