# Peer review of "Study of HgCdTe (100) and HgCdTe (111)B Heterostructures Grown by MOCVD and Their Potential Application to APDs Operating in the IR Range up to 8 µm"

_sensors, 2022, doi:10.3390/s22030924_

Round 1
Reviewer 1 Report
The authors explore high operating temperature avalanche photodiodes using HgCdTe. While there's much to like about the research area, the authors fail to connect their theoretical and experimental results. There are also multiple technical mistakes. I would urge the authors to take some time to clean up their technical mistakes, reduce their grammatical mistakes, and take a few more measurements before resubmitting. As such, the manuscript should be rejected or will require major revisions.
Here are some examples of the more egregious technical issues:
In Figure 3, the (III)B device has reduced responsivity with reverse bias. APDs need higher responsivity with reverse bias. The samples are not necessarily representative of APDs if they do not operate correctly. More evidence of their operation is needed to support the conclusions in this manuscript. Furthermore, there isn't any evidence that the devices are able to produce greater than 100% EQE, which is the proof of APD operation. It's also somewhat problematic that the two samples are not identical, including fairly large discrepancies in the p-type barrier layer.
In Figure 8, HgCdTe APDs are known to have F(M)<1.5. If so, why are they being mapped to the local field model?
Equation 2 is incorrect. The diffusion current somehow exists without a diffusion coefficient (D) or a mobility (mu) related through the Einstein equation (D = mu * k_B * T).
Equation 9 is missing a bandwidth term and a parenthesis. Unit analysis shows that the units will be in coulombs per rt second, not coulombs per second (amps). The parenthesis should be (Id + Iu)* M^2 * F. See Reference 7, equation 1.
Equation 10 is incorrect. It should be (2-1/M)*(1-k). And it's not clear why this equation is given. HgCdTe has lower noise than this equations permits.
Here are some examples of areas that could use improvement:
Why are the defects different between (111)B and (100) material? Researchers at SELEX have already indicated that GaAs (100) wafers with a slight miscut are preferable for APD development over GaAs (111)B wafers. This manuscript appears to support that claim, but without citing the original works or delving into the mechanisms. See Ian Baker (DOI: 10.1117/12.981850).
In Figure 5, the effects of BTBT and TAT aren't well represented. It might be better to have plots for just TAT and just BTBT.
In Figure 7, what is the assumed doping concentration? Obviously, this will impact whether or not the device can even deplete to 4 um.
It's easier to write "III-V" than AIIIBV. It's also more standard.
The standard notation for the impact ionization coefficients are alpha and beta, not alpha with subscripts. Alpha and beta are not described as field dependent, even though they are. This should be corrected.
There are several missing simulation parameters, including intrinsic carrier concentration and permittivity. These impact the dark current simulation and the depletion width. Please include your assumed values.

Author Response
We would like to thank the reviewer for comments, which undoubtedly influenced on improvements of final version of the paper. All the changes are highlighted in the revised manuscript. We suppose that all improvements will comply your requirements.
In Figure 3, the (III)B device has reduced responsivity with reverse bias. APDs need higher responsivity with reverse bias. The samples are not necessarily representative of APDs if they do not operate correctly. More evidence of their operation is needed to support the conclusions in this manuscript. Furthermore, there isn't any evidence that the devices are able to produce greater than 100% EQE, which is the proof of APD operation. It's also somewhat problematic that the two samples are not identical, including fairly large discrepancies in the p-type barrier layer.
Response: It should be noted that the analyzed photodiodes did not have the APD design, they were with PIN design. The aim of the work is to investigate which crystallographic orientation is more suitable for the design of an APD. The current responsivity of the (111)B photodiode decreases with the increase of the bias voltage due to the tunnel currents, which basically excludes its use for APD. For such preliminary analyzes, the level of doping of the p-type barrier is also irrelevant (the difference results from the properties of a given crystallographic orientation - the dopants activate in different ways, and therefore different concentration levels are achieved). If the energy gap in this area is extended, the dark current mainly comes from the absorber.
In Figure 8, HgCdTe APDs are known to have F(M)<1.5. If so, why are they being mapped to the local field model?
Response: For HgCdTe k values are of 0.1 or less, what gives F(M)<1.5. We do not know why used model is not correct, although it is found in many publications, incl. in the monograph M.A. Kinch, State-of-the-Art Infrared Detector Technology, SPIE Press, Bellingham, 2014].
Equation 2 is incorrect. The diffusion current somehow exists without a diffusion coefficient (D) or a mobility (mu) related through the Einstein equation (D = mu * k_B * T).
Response: The diffusion length is related to the minority carrier lifetime (and is determined by the mechanism with the shortest time). In our analyzes, we calculate the appropriate carrier lifetimes (radiative, Auger and SRH). The same model is used by M. Kinch [M.A. Kinch, State-of-the-Art Infrared Detector Technology, SPIE Press, Bellingham, 2014].
Equation 9 is missing a bandwidth term and a parenthesis. Unit analysis shows that the units will be in coulombs per rt second, not coulombs per second (amps). The parenthesis should be (Id + Iu)* M^2 * F. See Reference 7, equation 1.
Response: Equation and description have been corrected in the text.
Equation 10 is incorrect. It should be (2-1/M)*(1-k). And it's not clear why this equation is given. HgCdTe has lower noise than this equations permits.
Response: When electrons initiate the multiplication (and for HgCdTe in the analyzed range of molar composition, this is the case), F(M) can be calculated by F(M) =kM+(1-k)(2-1/M). In the literature, a notation as given in the text (this time in equation 13) is more common.
Here are some examples of areas that could use improvement:
Why are the defects different between (111)B and (100) material? Researchers at SELEX have already indicated that GaAs (100) wafers with a slight miscut are preferable for APD development over GaAs (111)B wafers. This manuscript appears to support that claim, but without citing the original works or delving into the mechanisms. See Ian Baker (DOI: 10.1117/12.981850).
Response: The source of the defects in HgCdTe and their influence on the dark currents is explained in the text. The differences in the currents between HgCdTe (100) and HgCdTe (111) B result mainly from different types of dislocations in the respective layers (unfortunately we do not have any research to prove it). Dislocations mainly affect the TAT, which depends on the defect density NT located at the energy level ET below the conduction band edge.
In Figure 5, the effects of BTBT and TAT aren't well represented. It might be better to have plots for just TAT and just BTBT.
Response: TAT depends on the defect density, NT. The reviewer is right that it would be better to show the graphs for TAT and BtBT separately, but showing TAT separately would make sense if we assumed a low dopant concentration and changed the defect concentration. In our calculations in Figure (5a) we assumed the constant NT, and changed the ND - in this case it is physically impossible that only TAT exist, without BtBT (for high ND).
In Figure 7, what is the assumed doping concentration? Obviously, this will impact whether or not the device can even deplete to 4 um.
Response: was added in the figure caption.
It's easier to write "III-V" than AIIIBV. It's also more standard.
Response: the notation in the text was standardized (using III-V notation).
The standard notation for the impact ionization coefficients are alpha and beta, not alpha with subscripts. Alpha and beta are not described as field dependent, even though they are. This should be corrected.
Response: In work [R.J. McIntyre, “A new look at impact ionisation-part I: a theory of gain, noise, breakdown probability, and frequency response”, IEEE Trans. Electron Devices 46(8), 1623-1631 (1999)] indeed the ionization coefficients for electrons and holes are denoted as alpha and beta, respectively. However, alpha notations with the appropriate subscript are also found. It is only a sign, if well described in the text, the reader should have no problem to distinguish it. In these phenomenological models, the electron (hole) ionization coefficient is assumed to depend only on electric field (F), with a form α = a×exp(−b/F ). a and b are parameters typical for given material, for HgCdTe we can use α = 3E×exp(−1.5E5/F ) cm−1 [p.111, M. A. Kinch, Fundamentals of Infrared Detector Materials, SPIE, Bellingham, 2007].
There are several missing simulation parameters, including intrinsic carrier concentration and permittivity. These impact the dark current simulation and the depletion width. Please include your assumed values.
Response: The missing parameters have been added to the table 2, including dependence on energy gap, intrinsic concentration, dielectric constants and effective masses.
Reviewer 2 Report
Overall, I think this manuscript warrants publication in Sensors journal. I find the following points to be of merit.
- The introduction covers why HgCdTe APDs have gained much interest in recent years.
 - A model is presented for expected noise factor in the HgCdTe APDs the authors used in their measurements.
 - Gain, dark current calculations seem to validate expected performance in low photon application regimes, and at high temperatures.

Perhaps the authors can make a comment on the following for the readers' benefit:
- Why not add an image showing the heterostructures grown by MOCVD? Is there any data such as XRR to show that the multilayers do not have issues such as rough interfaces?
 - HgCdTe APDs are already shown to be extremely capable in real world applications such as LiDARs - https://doi.org/10.1117/12.2599159. What advantages do the heterostructures developed by authors offer over those in others' work?

Author Response
We would like to thank the reviewer for comments, which undoubtedly influenced on improvements of final version of the paper. All the changes are highlighted in the revised manuscript. We suppose that all improvements will comply your requirements.
Why not add an image showing the heterostructures grown by MOCVD? Is there any data such as XRR to show that the multilayers do not have issues such as rough interfaces?
Response: As a standard, we do not perform XRD measurements of HgCdTe layers. For every few processes we perform SIMS measurements in order to verify the concentration level of elements, we also check interfaces with this method. Unfortunately, for these specific (analyzed in the article) layers, these measurements were not made. we can provide references that show the profiles of the layers obtained in our laboratory: P. Madejczyk, W. Gawron, P. Martyniuk, A. KÄ™bÅ‚owski, W. Pusz, J. Pawluczyk, M. Kopytko, J. Rutkowski, A. Rogalski, J. Piotrowski, “Engineering steps for optimizing high temperature LWIR HgCdTe photodiodes,” Infrared Physics & Technology 81, 276–281 (2017) [https://doi.org/10.1016/j.infrared.2017.01.020]
HgCdTe APDs are already shown to be extremely capable in real world applications such as LiDARs - https://doi.org/10.1117/12.2599159. What advantages do the heterostructures developed by authors offer over those in others' work?
Response: Classic IR APDs designed on the basis of homojunction must be cooled to the temperature of liquid nitrogen (as in the given paper) in order to reduce the mechanisms of thermal generation of carriers. The works indicate this, as the suggested work is up-to-date, we have also added it to the reference list. The use of heterostructure will make it possible to increase the detector operating temperature through non-equilibrium operating conditions enabling the suppression of Auger generation. We write in the text that we plan the diodes to be thermoelectrically cooled.
Reviewer 3 Report
Manuscript reports on comparative theoretical analysis of HgCdTe (100) and HgCdTe(111)B IR detectors with focus on the impact of the structural parameters of the multiplication region (its doping and thickness) on the dark currents and the gain parameters. The presented results are an extension of work reported by the authors and collaborators on similar topics in the past. The authors are the experts on this topic with good track of research papers.
In general, manuscript is written clearly, contains sufficient background which might be shorten for the sake of easy to read the Introduction (it is too wordy, make it up to the point, Table I is rather redundant). Furthermore, Dark Current and Avalanche Gain sections are very specific and contains textbook like materials which a potential reader would appreciate much more if authors established clearer link between equations and simulation results. Manuscript requires minor formatting and English proofreading to eliminate existing flaws.
It is trivial to mention here that composition of ternary or quaternary compounds such HgCdTe can be adjusted to get the target band gap, but the strain induced by lattice-mismatch must be carefully considered, otherwise the defects induced could cause a significant leakage current. This issue has not been considered in manuscript even though considered structure is heteroepitaxial. Why?
Furthermore, the predicted SRH carrier life-times for both considered HgCdTe structures resulting from a measurable NT density of mid band gap states need further substantiation and/or conformation. What is the physical origin of the mid band gap states for each structure, similarities, differences? Can authors provide HRTEM or other morphology analysis to support the predicted parameters (e.g. doi: 10.3390/nano11071855)? How to adjust relevant “level of growth technology” to address the identified issues affecting dark current and possible increase device operating temperature?
Author Response
We would like to thank the reviewer for comments, which undoubtedly influenced on improvements of final version of the paper. All the changes are highlighted in the revised manuscript. We suppose that all improvements will comply your requirements.
It is trivial to mention here that composition of ternary or quaternary compounds such HgCdTe can be adjusted to get the target band gap, but the strain induced by lattice-mismatch must be carefully considered, otherwise the defects induced could cause a significant leakage current. This issue has not been considered in manuscript even though considered structure is heteroepitaxial. Why?
Response: In the case of the ternary HgCdTe compound, the lattice constant change is not as large as in the case of the AIII-BV compounds (e.g., InAsSb). The lattice constant of HgTe is 0.646 nm, while the lattice constant of CdTe is 0.648 nm. Therefore, in this respect, HgCdTe offers the possibility to adjust the energy gap in a wide range, without changing the lattice constant – this is a significant advantage of this material over InAsSb.
Furthermore, the predicted SRH carrier life-times for both considered HgCdTe structures resulting from a measurable NT density of mid band gap states need further substantiation and/or conformation. What is the physical origin of the mid band gap states for each structure, similarities, differences? Can authors provide HRTEM or other morphology analysis to support the predicted parameters (e.g. doi: 10.3390/nano11071855)? How to adjust relevant “level of growth technology” to address the identified issues affecting dark current and possible increase device operating temperature?
Response: The source of the defects in HgCdTe and their influence on the dark currents is explained in the text. The differences in the currents between HgCdTe (100) and HgCdTe (111) B result mainly from different types of dislocations in the respective layers (unfortunately we do not have any research to prove it). Dislocations mainly affect the TAT, which depends on the defect density NT located at the energy level ET below the conduction band edge.